# Soybean Extracts (Glycine Max) with Curcuma, Boswellia, Pinus and Urtica Are Able to Improve Quality of Life in Patients Affected by CP/CPPS: Is the Pro-Inflammatory Cytokine IL-8 Level Decreasing the Physiopathological Link?

Tommaso Cai [1,2,*], Umberto Anceschi [3], Irene Tamanini [1], Paolo Verze [4] and Alessandro Palmieri [5]

1 Department of Urology, Santa Chiara Regional Hospital, 38123 Trento, Italy; irene.tamanini@apss.tn.it
2 Institute of Clinical Medicine, University of Oslo, 0010 Oslo, Norway
3 Department of Urology, IRCCS Regina Elena National Cancer Instituite, 00100 Rome, Italy; umberto.anceschi@gmail.com
4 Department of Urology, University of Salerno, 84121 Salerno, Italy; pverze@gmail.com
5 Department of Urology, University of Naples, Federico II, 80100 Naples, Italy; info@alessandropalmieri.it
* Correspondence: ktommy@libero.it; Tel.: +39-0461903306

**Abstract:** The present study evaluates the efficacy of a combination of soyabean extracts associated with Curcuma Longa, Boswellia, Pinus pinaster and Urtica dioica (PROSTAFLOG®) in patients affected by CP/CPPS, through the evaluation of interleukin-8 (IL-8) plasma seminal levels. All patients diagnosed with CP/CPPS, attending the same urologic center, were enrolled in this randomized, controlled phase III study. Participants were randomized to receive oral capsules of PROSTAFLOG® (two capsules at bedtime every 24 h) or Ibuprofen 600 mg (1 tablet daily), lasting for a period of four weeks. NIH-CPSI and SF-36 questionnaires, as urological evaluations with a transrectal ultrasound (TRUS), the Meares–Stamey test, and IL-8 dosage in seminal plasma were performed at baseline and at 3 months follow-up. A total of 77 patients (mean age of 34.5 ± 6.1) were enrolled (PROSTAFLOG® ($n$ = 39); ibuprofen ($n$ = 38)) in the study. At 3 months, in the PROSTAFLOG® series, 69.2% of patients showed a significant reduction in the NIH-CPSI score, compared with 34.2% in the ibuprofen group ($p < 0001$). The mean IL-8 levels were significantly lower in the PROSTAFLOG® cohort compared with the ibuprofen series ($p < 0.0001$), while a significant reduction in the IL-8 level between the enrolment and last follow-up evaluation was also observed in this group ($p < 0.0001$). Additionally, a significant reduction in the volume of the seminal vesicles assessed by TRUS was also found in the PROSTAFLOG® series during the observational timeframe (18.3 ± 7.1 mL vs. 11.2 ± 2.4 mL ($p < 0.0001$). In conclusion, PROSTAFLOG® significantly improves the QoL in patients affected by CP/CPPS and it provides a significant reduction in IL-8 seminal levels as the overall seminal vesicles volume.

**Keywords:** chronic pelvic pain; interleukin 8; chronic prostatitis; bromelain; soybean

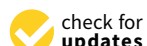



## 1. Introduction

Chronic prostatitis (CP) has been described as one of the most common diseases in men aged <50 year [1], with a significant impairment of the quality of life [2,3]. According to NIH classification [4], class III chronic prostatitis/chronic pelvic pain syndrome (CP/CPPS) has the largest incidence [5]. The optimal management of CP/CPPS represents an unmet need in urological practice as data regarding pathogenesis, evolution and treatment are scanty [6]. The historical treatment of CP/CPPS is based on the "three A's rule": antibiotics, anti-inflammatory medications and alpha blockers. In the era of bacterial resistance, the use of antibiotics for CP remains controversial, since urogenital samples of CP/CPPS patients are usually negative [7]. Phytotherapeutics represent an adequate alternative due to their negligible side-effects, despite the lack of general consensus and deducible

evidence from prospective controlled clinical trials [8–10]. In recent years, there has been an increased growing interest in several nutraceutical agents, such as glycine max [11], Boswellia [12], urtica dioica [13] and pinus pinaster [14], in the management of urinary tract infections (UTIs) and CP. For most of these compounds, the mechanism of action remains unclear. Nonetheless, several authors found that patients with CP/CPPS showed higher levels of pro-inflammatory cytokines, such as interleukin-8 (IL-8) [15,16]. Furthermore, IL-8 seems to be a promising biomarker for the evaluation of CP/CPPS severity with potential implications for diagnosis, prognosis, and treatment [17]. Our hypothesis is that phytotherapy can improve CP/CPPS patients' quality of life by reducing the IL-8 levels. In this context, we evaluate the efficacy and the safety of a combination of soyabean extracts (glycine max) associated with Curcuma Longa, Boswellia, Pinus pinaster and Urtica dioica (PROSTAFLOG®) in patients affected by CP/CPPS, by evaluating the role of the pro-inflammatory mediator (interleukin-8 (IL-8)) in CP/CPPS.

## 2. Materials and Methods

### 2.1. Study Design and Schedule

The study design was planned in accordance with the guidelines for clinical trials in CP/CPPS, described by the NIH Chronic Prostatitis Collaborative Research Network [18]. After internal board approval, between March 2020 and March 2021, all consecutive patients with a clinical and instrumental diagnosis of inflammatory CP/CPPS, attending the same urologic center, were enrolled in a randomized, controlled, not-blinded phase III study. The treatment group underwent (PROSTAFLOG®) (two capsules at bedtime every 24 h), while the control group was prescribed ibuprofen 600 mg (1 tablet daily). All patients received four weeks of treatment (28 day). No placebo arm was included. Attributable limitations, due to attrition biases pertaining to the knowledge of group assignments (not blinded), were acknowledged in the reporting of the outcomes. After the diagnosis of CP/CPPS, all eligible individuals signed a written informed consent and underwent baseline questionnaires, a urological examination using the Meares–Stamey test, transrectal ultrasound evaluation with the seminal vesicles volume measurement and collection of a seminal sample. Seminal vesicle plasma samples were used for IL-8 dosing. All patients who met the inclusion criteria were randomized through a computer-generated sequence of allocation, according to a 1:1 randomization. All patients were contacted by telephone on day 30 after starting therapy to ensure correct timing and dose of treatment. The first follow-up visit was scheduled at 3 months from starting therapy, with a urological and microbiological examination, questionnaire collection, transrectal ultrasound evaluation with the seminal vesicle volume measure and the IL-8 evaluation. Figure 1 summarizes the study schedule.

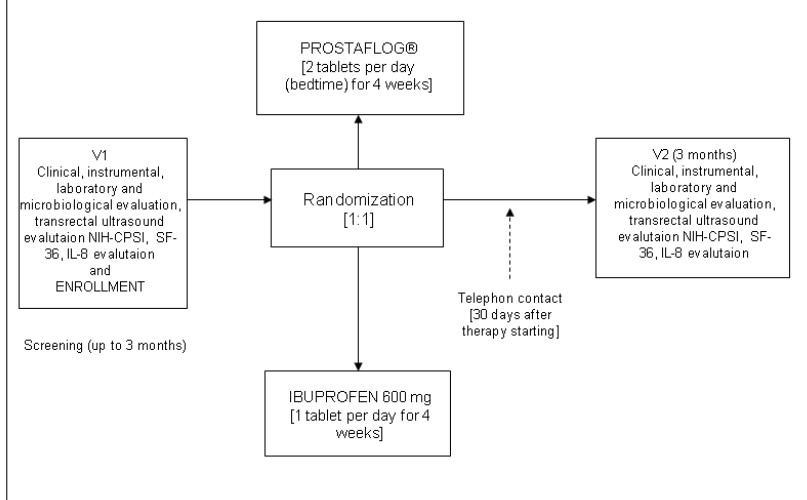

**Figure 1.** The study schedule.

### 2.2. Outcome Measures

The main outcome indicators were as follows: quality of life improvement, defined as improvement in the NIH-CPSI score (reduction in the NIH-CPSI total score by $\geq$25%) and decrease in IL-8 (reduction in at least 65% from the baseline value), at the end of the observational timeframe.

### 2.3. Inclusion and Exclusion Criteria

The inclusion criteria were the presence of symptoms of pelvic pain for at least three months during the six months before study entry, according to the European Association of Urology (EAU) guidelines, a score in the pain domain of the NIH-CPSI of >4 and a negative four-glass result in the Meares–Stamey test [19,20]. The exclusion criteria were as follows: male patients with <18 and >65 years of age, affected by major comorbidities and/or with known anatomical abnormalities of the urinary tract and/or with evidence of other urological diseases; patients affected by bladder outlet obstruction (BOO) with post-voiding residual urine volume >50 mL; patients with a reported allergy to pollen extract, who had recently (<4 weeks) undergone oral or parental treatment or who were currently using prophylactic antibiotic drugs; and all patients positive to tests for Chlamydia trachomatis, Ureaplasma urealyticum, Neisseria gonorrhoeae, Herpes Viruses (HSV 1/2) and Human Papillomavirus (HPV). We excluded all patients with >65 years of age and affected by bladder outlet obstruction (BOO) with a post-voiding residual urine volume >50 mL, in order to avoid the inclusion of patients with LUTS due to BPH.

### 2.4. Composition and Characterization of the Extracts Used

Each administration of PROSTAFLOG® contained Curcuma Longa 500 mg, boswellia 300 mg, Urtica dioica 240 mg, pinus pinaster 200 mg and glycine max 70 mg, as described in the manufacturer's instructions (Naturneed, Macerata, Italy). In the control group, all patients received ibuprofen 600 mg/die.

### 2.5. Questionnaires and Urological Examinations

Validated, self-administered Italian versions of the NIH-CPSI [21] and Short Form-36 (SF-36) questionnaires were administered to all patients [22] upon their first visit and at three months follow-up, respectively [23].

### 2.6. Microbiological Considerations, Sample Collection and Laboratory Procedures

All samples from the Meares–Stamey test and seminal plasma were collected at the time of the first urological visit and immediately brought to the laboratory, under refrigerated conditions, analyzed for cultures and aliquoted for DNA extraction and polymerase chain reaction for *Chlamydia trachomatis*, *Ureaplasma urealyticum*, *Neisseria gonorrhoeae*, HSV 1/2 and HPV detection. All subjects included in the study underwent urinary culture for common bacteria and yeasts. All microbiological evaluations and DNA extractions and purifications were carried out in accordance with the methods described by Mazzoli et al. [24]. Additionally, a natural human-produced IL-8 concentration was determined in the seminal samples of all the patients in both groups, by the solid-phase enzyme-linked immunosorbent assay (ELISA) Quantikine IL-8 (R&D Systems, Minneapolis, MN, USA) [24]. All samples were tested in duplicate by using independent analysis, in agreement with the manufacturer's recommendations and in order to avoid errors. The medium minimal detectable dose (MDD) of the IL-8 assay was 3.5 pg/mL [24].

### 2.7. Transrectal Seminal Vesicle Volume Measurement

The volume of the seminal vesicles was examined during the urological visit by a real-time ultrasound scanner with a rotating 7.5 MHz end-fire transducer (MyLab 70, Esaote S.p.A., Genova, Italy). All examinations were performed by the same experienced urologist. All patients underwent enema bowel preparation prior to the transrectal ultrasound scanner.

*2.8. Ethical and Statistical Considerations*

The study was conducted according to the Good Clinical Practice (GCP) guidelines, with the ethical principles established in the latest version of the Declaration of Helsinki. Written informed consent was obtained from all patients prior to the study enrolment. Furthermore, this trial was conducted with respect for the Consolidated Standards of Reporting Trials statement (The Ottawa Hospital Research Institute, Ottawa, ON, Canada). The homogeneity of the two groups at baseline was tested by Student's *t*-test and the Mann–Whitney U test for continuous variables, and $\chi^2$ test for categorical variables, respectively. General characteristics of the study participants were reported using descriptive statistics (means, standard deviations or interquartile ranges). Randomization based on a single sequence of random assignments (simple randomization) was performed using a pseudo-random number generator software (Research Randomizer Version 4.0, Social Psychology Network, Wesleyan University, Middletown, CT, USA). Analysis of variance (ANOVA) was used for means comparison. The Bonferroni adjustment test was also used at the second stage of the ANOVA. The differences between the groups, regarding the NIH-CPSI results and the mean concentration of pro-inflammatory cytokines, were also obtained using the ANOVA test. The calculation of the sample size to be enrolled was performed on the literature results, while the required sample size was calculated under the following conditions: the difference between groups, 35% of patients who reached a reduction of 25% of the NIH-CPSI total score; $\alpha$ error level, 0.05 two-sided; statistical power, 80%; and anticipated effect size, Cohen's d = 0.5. The calculation yielded $2 \times 32$ individuals per group. For all analyses, all *p*-values of <0.05 were considered statistically significant. Statistical analysis was performed using SPSS v22 (IBM Corp., Armonk, NY, USA).

## 3. Results

From an initial cohort of 134 patients attending our center in the enrolment study period, 81 met the inclusion criteria and were randomly allocated to the 40 PROSTAFLOG® group and 41 control series. Three patients in the ibuprofen cohort and 1 patient in the PROSTAFLOG® series were excluded for missing data during follow-up. In the per-protocol analysis, 77 males were eligible for analysis: 39 PROSTAFLOG® group and 38 control group. No statistically significant differences between the groups were identified. Moreover, no statistically significant difference was reported between the groups in terms of the IL-8 mean level at the baseline. All clinical and laboratory data at enrolment were summarized in Table 1.

**Table 1.** The demographic and clinical patient's data at the enrolment time (per protocol analysis).

| | PROSTAFLOG® Group | Ibuprofen Group | |
| --- | --- | --- | --- |
| | *Mean (SD \* or %)* | *Mean (SD \* or %)* | |
| Patients (n°) | 39 | 38 | |
| *Age* | 34.1 ± 5.2 | 34.4 ± 4.7 | 0.82 |
| *Educational qualification* | | | |
| Primary School | 3 (7.6) | 2 (5.2) | |
| High School | 26 (66.6) | 22 (57.8) | 0.65 |
| University | 10 (25.8) | 14 (37.0) | |
| *Sexual behaviour* | | | |
| 1 partner | 31 (79.4) | 32 (84.2) | |
| >1 partners | 8 (20.6) | 6 (15.8) | 0.43 |
| *Contraceptive use* | | | |
| Condom | 28 (71.7) | 26 (68.4) | |
| Coitus interruptus | 11 (28.3) | 12 (31.6) | 0.67 |
| *Start of CP/CPPS [#] history (months)* | 19.8 ± 4.9 | 19.5 ± 5.1 | 0.61 |

**Table 1.** *Cont.*

| | PROSTAFLOG® Group | Ibuprofen Group | |
|---|---|---|---|
| | *Mean (SD * or %)* | *Mean (SD * or %)* | |
| *Symptoms Score at baseline (mean) (range)* | | | |
| NIH-CPSI [§] | 25.8 ± 1.9 | 25.9 ± 1.7 | 0.43 |
| SF-36 [‡] | 93.6 ± 1.0 | 93.8 ± 1.1 | 0.49 |
| *Clinical presentation* | | | |
| Dysuria | 20 (51.2) | 18 (47.3) | |
| Urgency | 1 (2.4) | 2 (5.4) | 0.75 |
| Dysuria + Frequency | 8 (20.6) | 7 (18.4) | |
| Burning | 10 (25.8) | 11 (28.9) | |
| *Pain* | | | |
| Perineal | 15 (38.4) | 16 (42.1) | |
| Scrotal | 5 (12.8) | 4 (10.5) | 0.83 |
| Suprapubic | 10 (25.8) | 10 (26.3) | |
| Lower Abdominal | 9 (23.0) | 8 (21.1) | |
| *Pain frequency* | | | |
| Daily | 29 (74.2) | 30 (78.9) | 0.86 |
| Weekly | 10 (25.8) | 8 (21.1) | |
| *IL-8 mean level* (pg/mL) | | | |
| | 917 (332–>12,000) | 912 (418–>12,000) | 0.89 |

SD * = Standard Deviation; n° = number; NIH-CPSI [§] = NIH Chronic Prostatitis Symptom Index; SF-36 [‡] = Short Form-36; CP/CPPS [$] = chronic prostatitis/chronic pelvic pain syndrome.

### 3.1. Clinical and Instrumental Results at the Follow-Up Evaluation

At 3 months follow-up, in the PROSTAFLOG® group, 27/39 patients (69.2%) showed a significant reduction in the NIH-CPSI total score (more than 25% from baseline), compared with 13/38 (34.2%) in the ibuprofen group ($p < 0.0001$). The SF-36 questionnaires showed a statistically significant difference between groups (SF-36 mean value PROSTAFLOG®: 97.9 ± 1.8 vs. SF-36 mean value ibuprofen group: 94.6 ± 3.9 ($p < 0.001$)). A detailed report of all the questionnaire results at three months after treatment is described in Table 2. Furthermore, a statistically significant reduction in the total volume of the seminal vesicles at the transrectal ultrasound evaluation was found in the PROSTAFLOG® group between the baseline and follow-up visit (18.3 ± 7.1 mL vs. 11.2 ± 2.4 mL ($p < 0.0001$)).

### 3.2. Laboratory Results at the Follow-Up Evaluation

The mean levels of IL-8 were significantly lower in the PROSTAFLOG® group compared with the ibuprofen series (IL-8 301 pg/mL vs. IL-8 527 pg/mL ($p < 0.0001$)). In the PROSTAFLOG® cohort, we also found a statistically significant reduction in the IL-8 level between the enrolment and last follow-up visit (IL-8 917 pg/mL vs. IL-8 301 pg/mL ($p < 0.0001$)). The mean level of IL-8 at the last follow-up evaluation is detailed in Table 2. All patients were negative at the Meares–Stamey test evaluation.

### 3.3. Adverse Effects

A total of 2 patients out of 39 (5.1%) in the PROSTAFLOG® group and 8/38 patients (21%) in the control series, reported mild adverse effects (mild nausea) that did not require any additional treatment.

**Table 2.** The clinical and laboratory results at the follow-up evaluation.

| | PROSTAFLOG® Group | Ibuprofen Group | |
|---|---|---|---|
| | *Mean (SD *)* | *Mean (SD *)* | |
| **NIH-CPSI °** | | | |
| *pre-treatment* | 25.8 ± 1.9 | 25.9 ± 1.7 | 0.43 |
| *post-treatment* | 10.8 ± 2.8 | 21.7 ± 4.1 | <0.001 |
| *p* | <0.001 | 0.0003 | |
| **(NIH-CPSI pain domain)** | | | |
| *pre-treatment* | 11.5 ± 1.9 | 11.1 ± 2.6 | 0.31 |
| *post-treatment* | 6.6 ± 1.4 | 8.8 ± 2.2 | 0.003 |
| *p* | <0.001 | <0.001 | |
| Reduction of NIH-CPSI pain domain | −5.4 ± 0.2 | −3.0 ± 0.5 | <0.001 |
| **SF-36 ‡** | | | |
| *pre-treatment* | 93.6 ± 1.0 | 93.8 ± 1.1 | 0.91 |
| *post-treatment* | 97.9 ± 1.8 | 94.6 ± 3.9 | <0.001 |
| *p* | <0.001 | 0.13 | |
| **IL-8 mean level (pg/mL)** | | | |
| *pre-treatment* | 917 (332–>12,000) | 912 (418–>12,000) | 0.89 |
| *post-treatment* | 301 (<100–3460) | 527 (346–>12,000) | <0.001 |
| *p* | <0.001 | 0.003 | |
| **Seminal vesicles total volume (mL)** | | | |
| *pre-treatment* | 18.3 ± 7.1 | 19.1 ± 6.2 | 0.77 |
| *post-treatment* | 11.2 ± 2.4 | 15.9 ± 9.1 | <0.001 |
| *p* | <0.001 | 0.09 | |

SD * = Standard Deviation; NIH-CPSI ° = NIH Chronic Prostatitis Symptom Index; SF-36 ‡ = Short Form-36.

## 4. Discussion

### 4.1. Major Finding

To our knowledge, this is the first study that demonstrated the clinical efficacy and safety of PROSTAFLOG® in the management of patients affected by CP/CPPS. Furthermore, we also described a significant association between IL-8 reduction and its clinical response. Finally, we found that PROSTAFLOG® provides a significant reduction in the overall seminal vesicle volume. The critical reduction of the seminal plasma IL-8 level and seminal vesicle overall volume should be considered as potential clinical markers to evaluate a significant response to drug treatment.

### 4.2. Results in Comparison with Other Studies

Boswellia has been traditionally considered as a potential therapeutic option against several chronic inflammatory diseases, such as chronic prostatitis and chronic pelvic pain syndrome. Moreover, Sibona et al. recently demonstrated the clinical efficacy of a phytotherapeutic agent containing boswellia in the management of patients affected by prostatitis-like symptoms, showing a significant enhancement of symptoms relief [25]. Similarly, several authors reported clinical benefits of a curcumin supplementation on main functional parameters related to prostatic diseases [26,27], such as pain relief [27]. Curcumin, a bioactive polyphenol from turmeric, is a common anti-inflammatory agent in preclinical research, and a recent systematic review demonstrated its anti-inflammatory effects, suggesting further investigations to confirm dose, duration and formulation to optimize its implementation in urological practice for chronic inflammation of the lower urinary tract [28]. Finally, several authors showed the beneficial effects of soybean extracts (glycine max) on the prostate physiology [29], as well as the benefits of pinus pinaster [30]. To date, the effect of all these compounds included in a single phytotherapeutic administration on the symptoms relief and reduction in pro-inflammatory cytokines has never been explored. To our knowledge,

this is the first report describing the clinical effects of PROSTAFLOG® associated with the reduction in IL-8 levels and the overall total volume of seminal vesicles. In recent years, Penna et al. demonstrated that among all the profiles of cytokines and chemokines, IL-8 appears to be the most reliable and predictive immunologic marker to identify inflammatory conditions of prostate, such as CP/CPPS and BPH15. Since neither biological nor molecular markers for evaluating treatment response are currently available for CP/CPPS, our findings highlight the role of IL-8 as a novel indicator in this complex clinical scenario. Finally, as the relationship between higher seminal vesicle volume and prostatitis-like symptoms remains traditionally unclear, herein we demonstrated that a significant reduction in the overall volume of seminal vesicles may be obtained by PROSTAFLOG® administration. This novel compound is associated with a decrease in symptoms, significant IL-8 reduction and the improvement of quality of life. IL-8 represents a potential marker for predicting a significant clinical response to treatment in patients affected by CP/CPPS, and further evidence is eagerly awaited.

*4.3. Strengths and Limitations of the Present Study*

The current study identified the following key factors to consider when treating CP/CPPS: the evaluation of seminal vesicles' IL-8 levels, seminal vesicles' overall volume and their implications on clinical results. Nonetheless, this pilot study had several limitations that should be acknowledged: the small sample size and observational timeframe, and a restricted patient population as the attrition bias. Moreover, the lack of long-term follow-up results and the long-term adverse effects should be taken into account.

**5. Conclusions**

PROSTAFLOG® treatment may significantly improve QoL in patients affected by CP/CPPS, providing a significant decrease in IL-8 and overall seminal vesicles' total volume, respectively. A critical reduction in IL-8 may be eventually considered as a prognostic factor of clinical response. Considering the poor efficacy of conventional therapies, alternative options for the management of CP/CPSS represents an unmet need in urological practice. In this scenario, PROSTAFLOG® may represent a promising, safe and noteworthy therapeutic agent. Further evidence form larger and prospective series is needed to clearly establish the role of PROSTAFLOG® in the management of CP/CPSS in the long run.

**Author Contributions:** Conceptualization, T.C.; data collection, T.C., I.T. and U.A.; writing—original draft preparation, T.C. and U.A.; writing—review and editing, P.V. and A.P. All authors have read and agreed to the published version of the manuscript.

**Funding:** This research received no external funding.

**Institutional Review Board Statement:** Ethical review and approval were waived for this study, due to the fact that according to the Italian regulations the phytotherapeutic compounds did not require the Ethical approval by the Ethics Committee. However, the study was conducted according to the guidelines of the Declaration of Helsinki.

**Informed Consent Statement:** Even if not required by the Italian regulations on the phytotherapeutic compounds, written informed consent has been obtained from the patients to publish this paper.

**Conflicts of Interest:** The authors declare no conflict of interest.

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
