# Peer review of "Soybean Extracts (Glycine Max) with Curcuma, Boswellia, Pinus and Urtica Are Able to Improve Quality of Life in Patients Affected by CP/CPPS: Is the Pro-Inflammatory Cytokine IL-8 Level Decreasing the Physiopathological Link?"

_2673-4397, doi:10.3390/uro2010006_

Round 1

Reviewer 1 Report

This paper is well done.

I very much appreciated your paper.

Author Response

Many thanks for your comment.

Reviewer 2 Report

This study on 77 patients with Chronic Prostatitis/ Chronic Pelvic pain syndrome shows an improvement in Quality of life and reduction of inflammatory cytokine IL-8 in seminal plasma with the use of Soybean extracts (Glycine Max) with Curcuma, Boswellia, Pinus, and Urtica.

I would like to congratulate the authors for their efforts. However, I have certain comments.

Introduction: Please add your hypothesis in 1-2 lines at the end of this section.

Materials and methods:

-What was the reason to exclude patients more than 65 years of age?
-Please mention the reason to exclude patients with bladder outlet obstruction
-In lines 110 and 111, it was mentioned, all investigations
were done at 1st visit. For TRUS, did you use any bowel
preparation?

Results

-In line 169, it was misprinted as [18.3±7.1 ml vs IL-8
11.2±2.4 ml] instead of [18.3±7.1 ml vs 11.2±2.4 ml]. Please correct this.
-Line 182- no mention of adverse effects in either group. Please mention.

Discussion:

-No mention of long-term follow-up, how long to
continue regime, results after stopping the medication,
adverse effects with long-term use. Please discuss these.
-Please discuss the limitations of this study in detail.

Author Response

Many thanks for your comments and suggestions.

In line with your suggestion, the following sentence has been added to the Introduction: "Our hypothesis is that phytotherapy is able to improve CP/CPPS patients’ quality of life by reducing the IL-8 levels".

We excluded all patients aged more than 65 years in order to avoid an overlap between CP/CPPS and LUTS due to BPH. In line with your suggestions, the following sentence has been added to the M&M section: "We excluded all patients with  >65 years of age and affected by bladder outlet obstruction (BOO) with post-voiding residual urine volume >50 ml in oder to avoid to include patients with LUTS due to BPH".

In line with your suggestion, the following sentence has been added to the M&M section: "All patients underwent enema bowel preparation prior to transrectal ultrasound scanner".

The Results section has been improved as suggested.

The Discussion section has been improved as suggested.

Thanks a lot, again, for your comments.